# Associations of Dietary Zinc–Vitamin B6 Ratio with All-Cause Mortality and Cardiovascular Disease Mortality Based on National Health and Nutrition Examination Survey 1999–2016

**DOI:** 10.3390/nu15020420

**Published:** 2023-01-13

**Authors:** Naijian Zhang, Zhilin Li, Qingcui Wu, Huijie Huang, Siting Wang, Yuanyuan Liu, Jiageng Chen, Jun Ma

**Affiliations:** 1School of Public Health, Tianjin Medical University, Tianjin 300070, China; 2Tianjin Key Laboratory of Environment, Nutrition and Public Health, Tianjin 300070, China

**Keywords:** dietary zinc, dietary vitamin B6, cardiovascular disease, all-cause mortality, restricted cubic spline

## Abstract

Previous studies have suggested a possible association among dietary zinc and vitamin B6 intake and CVD mortality and all-cause mortality. However, evidence on the association of dietary zinc and vitamin B6 intake and their interactions with CVD mortality and all-cause mortality remains unclear. This prospective study utilized data from the US National Health and Nutrition Examination Survey (NHANES) from 1999 to 2016. After a median follow-up of 10.4 years, 4757 deaths were recorded among 36,081 participants. Higher dietary zinc intake levels (≥9.87 mg/day) were associated with lower CVD mortality (hazard ratio (HR) = 0.85, 95% confidence interval (CI): 0.83–0.87). Vitamin B6 intake levels (≥1.73 mg/day) were associated with lower CVD mortality (HR = 0.91, 95% CI: 0.86–0.96) and all-cause mortality (HR = 0.91, 95% CI: 0.90–0.93). Higher dietary zinc intake and higher vitamin B6 intake were associated with a lower risk of CVD mortality, with an interaction between dietary zinc intake levels and vitamin B intake levels (LZLV group (HR, CI): 1.21,1.12–1.29; LZHV group (HR, CI): 1.42, 1.34–1.50; LZHV group (HR, CI): 1.28, 1.14–1.45; HZHV group (HR, CI): ref). There was also a J-type association (*p* for nonlinear < 0.001) between the dietary zinc–vitamin B6 ratio and CVD mortality, with a high dietary zinc–vitamin B6 ratio increasing the risk of CVD mortality (HR = 1.27, 95% CI: 1.19–1.35), whereas a moderate dietary zinc–vitamin B6 ratio appeared to be beneficial for CVD mortality. These results suggest that increasing the appropriate proportion of dietary zinc and vitamin B6 intake is associated with a lower risk of CVD mortality. Furthermore, precise and representative studies are needed to verify our findings.

## 1. Introduction

Cardiovascular disease (CVD) is a disease of the heart and blood vessels characterized by coronary heart disease, angina pectoris, congestive heart failure, myocardial infarction, and stroke [1]. CVDs, a primary cause of death, are a rapidly growing public health problem worldwide, with more people dying each year from CVDs than from any other cause [2]. CVD deaths are projected to grow annually from 17.6 million in 2016 to 23.6 million by 2030 [3]. In previous studies, the risk of developing CVD has been found to be influenced by lifestyle, smoking, poor diet, and lack of physical activity [4]. Therefore, dietary intervention for CVD has become an important preventative and therapeutic topic.

Zinc is an essential micronutrient in the diet that is involved in a variety of protein–protein interactions, fatty acid metabolism, apoptosis. and signal transduction and has three main biological roles, namely, acting as a catalyst, structural ion, and regulatory ion [5]. In humans, severe zinc deficiency has serious implications for growth retardation, neuronal dysfunction, and anemia; marginal zinc deficiency is associated with various pathological conditions, particularly age-related diseases including CVD [6]. In addition, zinc deficiency can affect liver disease [7,8] and diabetes [9,10]. However, in subsequent studies, the results regarding the relationship between dietary zinc and CVDs have been inconsistent. A study of Australian women (50–61 years) found a positive association between dietary zinc intake and CVD risk [2]. Some reports have also shown a negative association between dietary zinc intake and CVD risk [11]. In previous studies, the relationship between dietary zinc and CVD risk has rarely been investigated with large national samples. It has been suggested that in addition to being associated with inadequate intake of total dietary zinc, the etiology of zinc deficiency is associated with limitations in zinc bioavailability based on the dietary zinc type [12,13]. Yet, only dietary zinc intake has been considered in many studies.

In an experiment exploring the effect of vitamin B6 on zinc absorption in rats, it was found that increasing levels of vitamin B6 in the daily diet increased zinc absorption. Zinc absorption was lowest in rats fed 2 ppm vitamin B6 and highest in rats fed 40 ppm vitamin B6 [14]. A significant negative association between dietary intake of vitamin B6 and mortality and morbidity from CVD has been found in past studies [15]. To our knowledge, there is limited evidence on the overall effect of dietary zinc intake or its interaction with vitamin B6 on CVD mortality. No previous studies have investigated the effect of the interaction between dietary zinc and vitamin B6 on CVD mortality and all-cause mortality and the proportion of intake associated with all-cause mortality and CVD. This prospective study hypothesized that dietary zinc intake and vitamin B6 intake are associated with CVD and all-cause mortality outcomes.

Therefore, the association of single dietary intake of zinc or vitamin B6, the interaction of dietary zinc with vitamin B6, and intake of the zinc–vitamin B6 ratio with all-cause mortality and CVD mortality in the National Health and Nutrition Examination Survey (NHANES, 1999–2016) were assessed.

## 2. Materials and Methods

### 2.1. Study Design and Population

NHANES is a sample survey conducted by the National Center for Health Statistics of the Centers for Disease Control and Prevention to assess the health of adults and children in the United States. NHANES uses a complex, multi-stage probability sampling design with approximately 5000 individuals sampled each year [16]. The strength of the survey is that it combines a physical examination with an interview. Interviews that included questionnaires on demographic, socio-economic, diet, and health-related issues were administered and collected by trained interviewers. Physical and laboratory examinations were also conducted by trained medical professionals at mobile examination centers [17]. The present investigation includes nine two-year cycles of NHANES data from 1999 through 2016. A unique identifier (SEQN) was used for each respondent in the data file in question. A total of 36,081 participants were included in this study. We excluded (a) 38,807 respondents who lacked follow-up; (b) 149 respondents whose age at the time of death was less than 35 years; (c) 5803 respondents whose dietary zinc/vitamin B6 intake was 0 or deficient; (d) 3573 respondents who reported total energy intake exceeding predetermined limits (<500 or >3500 kcal/day for women and <800 or >4000 kcal/day for men) or total energy intake deficit [18]; (e) 7641 respondents who had been diagnosed with congestive heart failure/coronary artery disease/angina/heart attack/stroke; (f) 8 respondents with missing sample weights. The detailed exclusion criteria for inclusion are shown in Figure 1.

### 2.2. Ethics

The NHANES protocol was approved by the ethical review committee of the National Centre for Health Statistics Research, and all participants signed informed consent. NHANES presents details via the web (www.cdc.gov/nchs/nhanes/index.htm; accessed on 1 December 2022)

### 2.3. Dietary and Exposure Assessment

In the NHANES survey, trained enumerators measured food intake through a 24 h dietary recall interview on two consecutive days. In the 1999 to 2002 surveys, only one dietary retrospective interview over 24 h was conducted at the mobile examination centers. A second interview was added by telephone in 2003 to 2015. The 2nd dietary recall interview was conducted by telephone 3–10 days later. Dietary nutrients and energy intake were coded, and nutrient values were determined using the USDA Food and Nutrient Database (FNDDS). The mean of nutrient intakes from day 1 and day 2 of the 24 h dietary recall interview were used in the analysis. Using dietary intake data from two nonconsecutive days is a more accurate estimation method than using data from a single day [19].

### 2.4. Ascertainment of Incident Hypertension

The results of this study are cardiovascular cause deaths and deaths of any cause occurring before December 31, 2019, linking NHANES data to the National Death Index (NDI) death certificate records. Causes of death were defined using the International Statistical Classification of Diseases and Related Health Problems, 10th Revision (ICD-10), ICD-10 codes: I00–I09, I11, I13, I20–I51.

### 2.5. Other Covariates

In the NHANES survey, demographic and lifestyle variables were obtained using a standardized questionnaire. Age, sex, education, ethnicity, family income–poverty ratio, alcohol consumption, smoking, glycosylated hemoglobin, presence of diabetes, presence of hypertension, and presence of high cholesterol were used as covariates. Data were subdivided based on sex (male or female), body mass index (BMI) calculated as body weight in kilogram divided by squared height (kg/m^2^; BMI < 30 or BMI ≥ 30), educational attainment (below high school, high school or equivalent, college and above), race/ethnicity (Mexican American, non-Hispanic white, non-Hispanic black, other), family income–poverty ratio (<1.0, 1.0–3.0, >3.0), current smoker (never, ever or currently, smoked at least 100 cigarettes during their lifetime but not currently smoking, defined as abstinent), current drinker (≥12 alcoholic beverages per year), glycated hemoglobin A1c (HbA1c; <7% or ≥7%), self-reported hypertension (yes or no), and self-reported hypercholesterolemia (yes or no).

### 2.6. Statistical Analysis

All analyses in this study included sample weights, clustering, and stratification to account for the complex sampling design of NHANES data, in accordance with the NHANES analysis guidelines. Descriptive analysis was conducted for demographic characteristics between the two groups classified by median total dietary zinc. Selected demographic characteristics and biochemical measures were expressed as mean ± standard deviation of continuous variables and percentage (%) for categorical variables, respectively. Chi-square tests were used to analyze categorical variables. *t*-tests were used to analyze normal continuous variables.

A multivariate Cox proportional risk regression model was used to calculate risk ratios (HRs) and 95% CIs for dietary zinc and vitamin B6 levels in relation to CVD risk and all-cause mortality. The proportional risk assumption was tested using the KM survival curve, and the results showed that the proportional risk assumption was not violated. Groups were divided according to dietary zinc and median dietary vitamin B6 levels. The first group was set as a reference group that was used to obtain the low and high groups. Two multivariate models were developed. Model 1 was adjusted for age (continuous, years) and gender (male or female). Model 2 was further adjusted for race and ethnicity (Mexican American/non-Hispanic White/non-Hispanic Black/other), education level (<high school, high school or equivalent, or college), household income to poverty ratio (<1.0, 1.0–3.0, or >3.0), BMI (<30.0 or ≥30.0 kg/m^2^), smoking status (never, ever, or current), drinking status (yes/no), HbA1c (<7% or ≥7%), self-reported history of hypertension (yes or no), and self-reported high cholesterol history of hypertension (yes or no).

Participants were divided into four groups based on median dietary zinc and vitamin B6 intake: low zinc, low vitamin B6 score (LZLV) group; low zinc, high vitamin B6 score (LZHV) group; high zinc, low vitamin B6 score (HZLV) group; and high zinc, high vitamin B6 score (HZHV). Using the HZHV group as a reference, a multi-factor Cox proportional risk regression model was used to assess the risk of the other three groups separately. The *p*-value of the product term between dietary zinc and vitamin B6 levels was used to estimate the significance of the interaction. The dietary zinc and vitamin B6 ratios were log-transformed prior to analysis. Between the 0.5th and 99.5th percentile of dietary zinc and vitamin B6 ratios, restricted cubic spline analysis in three sections (10th, 50th, 90th percentile) was used to examine the non-linear relationship between dietary zinc levels and vitamin B6 levels and CVD mortality and all-cause mortality. Schoenfeld residuals were used to test for proportional risk hypothesis, and no violations were observed.

The quartiles of the log-transformed dietary zinc and vitamin B6 ratios were divided into four groups. The second and third quartiles were combined into one reference group to obtain the low (quartile 1), medium (quartiles 2 and 3), and high (quartile 4) groups. Cox proportional risk models were used to estimate the risk ratio (HR) and 95% confidence interval (CI) for all-cause mortality and CVD mortality for each standard deviation increase in the ratio of dietary zinc to vitamin B6. Dietary zinc (high/low) and vitamin B6 (high/low) were also added to the adjustment factors in Model 2 to eliminate the effect of dose on the outcome.

A series of sensitivity analyses were carried out in this study. First, participants who died within 2 years of follow-up were excluded. Second, permissible total energy intake limits were set to percentiles, 1 to 99. Finally, additional adjustment was made for a history of diabetes and for dietary fiber intake.

A two-sided *p* < 0.05 was set as the threshold for statistical significance. All analyses were performed using R version 4.1.3 (R Foundation, https://www.r-project.org/ (accessed on 11 December 2022)).

## 3. Results

### 3.1. Subsection

During a median follow-up period of 10.4 years, of 36,081 participants (mean age 45.63 (16.36) years; 16,491 men (weighted, 45.7%) and 19,590 women (weighted, 54.3%)), 4757 participants died, including 1063 from CVD and 3694 from other diseases. Participants with above-median dietary zinc intake levels were significantly younger, tended to be male and current drinkers, had higher rates of education, had a higher family income–poverty ratio, were former smokers, and had a lower prevalence of being told they had hypertension than participants with below-median dietary zinc intakes. There were no significant differences in BMI, HbA1c, or the prevalence of participants being told they had high cholesterol. Compared to participants with below median dietary vitamin B6 intake, those with above median dietary vitamin B6 intake were significantly younger, tended to be male and current drinkers, had a higher education, a higher family income–poverty ratio, a higher proportion of current non-smokers, and a lower prevalence of high BMI (≥30) and HbA1c (≥7%), and a lower prevalence of being told they had high blood pressure. There was no significant difference in the prevalence of participants being told they had high cholesterol (Table 1).

### 3.2. Dietary Zinc and Vitamin B6 Intake Levels in Association with CVD Mortality and All-Cause Mortality

After multifactorial adjustment, the HR (95% CI) for CVD mortality in the high dietary zinc intake group was 0.85 (0.83–0.87) compared with the reference group (low dietary zinc intake group), the HR (95% CI) for CVD mortality in the high vitamin B6 intake group was 0.91 (0.86–0.96) compared with the reference group (low vitamin B6 intake group). The HR (95% CI) for all-cause mortality in the high dietary zinc intake group was 0.93 (0.87–1.00) compared with the reference group (low dietary zinc intake group), and the HR (95% CI) for all-cause mortality in the high vitamin B6 intake group was 0.91 (0.90–0.93) compared with the reference group (low vitamin B6 intake group) (Table 2). Mortality and vitamin B6 intake levels were negatively associated with all-cause mortality. These results suggest that higher dietary zinc intake levels reduce the risk of CVD mortality, and higher vitamin B6 intake levels reduce the risk of CVD mortality and all-cause mortality.

### 3.3. The Association of Dietary Zinc Intake and Vitamin B6 Interactions with CVD Mortality and All-Cause Mortality

The association of the interaction between dietary zinc and vitamin B6 intake with CVD mortality was statistically significant. The LZLV, LZHV, and HZLV groups had an increased risk of CVD mortality compared to the HZHV group, with HRs (95% CI) of 1.21 (1.12–1.29), 1.42 (1.34–1.50), and 1.28 (1.14–1.45), respectively. The interaction of dietary zinc and vitamin B6 intake on all-cause mortality was not statistically significant. The risk of all-cause mortality was increased in the LZLV and LZHV groups compared to the HZHV group, with HRs (95% CI) of 1.12 (1.07–1.17) and 1.05 (1.00–1.10) (Table 3).

### 3.4. Dietary Zinc–Vitamin B6 Ratio and Risk of All-Cause and CVD Mortality

In the Cox model, restricted cubic splines showed a J-shaped relationship between the dietary zinc–vitamin B6 ratio and CVD mortality and all-cause mortality (*p* nonlinear < 0.001) (Figure 2). After adjusting for multiple factors in Model 2, the HR (95% CI) for CVD mortality was 1.00 (0.68–1.46) for the first cohort and 1.27 (1.19–1.35) for the third cohort, compared with the reference group (second and third quartiles) (Table 4).

### 3.5. Sensitivity Analyses

Samples were stratified by the LZLV group, LZHV group, HZLV group, and HZHV group. In each stratum, the hazard ratio increased with increasing dietary zinc–vitamin B6 ratio when the dietary zinc–vitamin B6 ratio exceeded a certain level (Appendix A).

Multiple sensitivity analyses of the multivariable-adjusted positive association between dietary zinc–vitamin B6 ratio and CVD mortality were performed. To minimize potential reverse causality bias, participants who died within 2 years of follow-up were excluded. A percentile from 1 to 99 of permissible total energy intake limit was used. Models were additionally adjusted for history of diabetes mellitus and for dietary fiber intake. Results were robust in sensitivity analyses (Table 5).

## 4. Discussion

From this large prospective cohort study of a nationally representative sample of US adults, an association was shown between reduced CVD mortality and increased dietary zinc intake (≥9.87 mg/day). A vitamin B6 intake level ≥ 1.73 mg/day was associated with lower CVD mortality and all-cause mortality. The results of the interaction analysis showed that higher dietary zinc intake and higher vitamin B6 intake were associated with a lower risk of CVD mortality and that there was an interaction between dietary zinc intake levels and vitamin B intake. Moreover, after adjusting for potential CVD confounders and dietary zinc and vitamin B6 intakes, there was also a J-type association between dietary zinc–vitamin B6 ratios and CVD mortality, with high dietary zinc–vitamin B6 ratios increasing the risk of CVD mortality, while moderate dietary zinc–vitamin B6 ratios appeared to be beneficial for CVD mortality. Sensitivity analyses suggest that these findings are robust. To our knowledge, this study is the first to provide direct evidence of the interaction between dietary zinc and vitamin B6 with CVD mortality and the proportional relationship between zinc and vitamin B6 intake and CVD mortality.

The association between dietary zinc levels and CVD mortality has been examined in different populations, and the results are not uniform. In a 1999–2010 NHANES study, adequate nutritional intake of zinc (RR = 0.50, 95% CI: 0.36–0.71) was associated with lower CVD mortality [20]. In a multi-racial study of 5186 participants (48.8% male; 41.3% white, 25.0% black, 21.6% Hispanic, and 12.1% Chinese American) in a fully adjusted model, higher dietary zinc was associated with a lower risk of coronary artery calcium progression in men (HR = 0.697, 95% CI: 0.553–0.878; *p* = 0.002) and women (HR = 0.675; 95% CI 0.496–0.919; *p* = 0.012, both compared to the extreme group) were associated with a lower risk of coronary artery calcium progression [11]. Among healthy Japanese men and women aged 40–79 years, higher zinc intake was negatively associated with coronary heart disease mortality in men (HR = 0.68, 95% CI: 0.58–1.03; *p* trend = 0.05) but not in women (HR = 1.13, 95% CI: 0.71–1.49; *p* trend = 0.61) [21]. An assessment of dietary intake in 466 male participants at the Shahid Gangalal National Heart Centre found a significant inverse association between dietary zinc intake equal to or exceeding the recommended daily intake and coronary artery disease (OR in G-estimation method: 0.91; 95% CI: 0.87, 0.96; OR in IPTW: 0.73; 95% CI: 0.66, 0.82) [22]. In a study of postmenopausal women aged 55–69 years (*n* = 34,492), dietary zinc intake was not associated with the risk of CVD mortality; however, the corresponding RRs for dietary zinc were 1.0, 0.61, 0.59, 0.57, and 0.37 among drinkers with 10 g/day of alcohol consumption (*p* for trend 0.07). In an analysis restricted to those drinking 30 g/day of alcohol, the risk gradient was enhanced [13]. In 27,742 person-years of follow-up (70 CVD deaths), no association was observed between CVD deaths and dietary zinc levels [23]. Among middle-aged Australian women (50-61 years), those in the highest quintile of zinc intake were more likely to have cardiovascular disease compared to those in the lowest quintile of zinc intake (OR = 1.67, 95% CI = 1.08–2.62) [2]. These inconsistent findings may partly be due to small sample sizes or that dietary zinc levels alone do not represent a complete explanation for the overall change in CVD mortality. Zinc levels may need to be studied in conjunction with vitamin B6 levels. In the current study with a larger sample size, we found that dietary zinc levels in 36,081 US adults were associated with CVD and all-cause mortality, and that lower dietary zinc intake (<9.89 mg/day), after adjusting for potential confounders, was associated with a higher risk of CVD.

Increased inflammation and impaired endothelial cell structure may explain the effects of zinc deficiency on atherosclerosis, with several studies reporting that zinc deficiency leads to the release of atherogenic factors that subsequently lead to the development of CVD [24]. A zinc-deficient diet can also lead to oxidative stress, which can contribute to the development of hypertension [6]. In 2019, the results of a study on mice which assessed the effect of dietary zinc levels on blood pressure showed that mice fed a zinc-deficient diet had increased blood pressure [25]. Hypertension is known to be a risk factor for CVD [26,27]. Zinc deficiency may be an indirect risk factor for cardiovascular disease but current findings are inconsistent, so the exact clinical benefits and dosage range of dietary zinc intake to prevent CVD have not been determined.

In a Japanese cohort of 23,119 men and 35,611 women, vitamin B6 intake was found to be negatively associated with mortality from heart failure in men and mortality from stroke, coronary heart disease, and total CVD in women [28]. In a hospital-based case-controlled study of 419 patients with Type 2 diabetes (T2D) newly diagnosed with CVD and age- (±5 years) and sex-matched T2D patients, vitamin B6 was found to be associated with a lower risk of CVD in patients with T2D [29]. In a study of 9142 participants in Korea, higher dietary intake levels of vitamin B6 were found to be associated with a reduced risk of CVD in Korean men [30]. In the current study of two prospective Chinese cohorts, high dietary vitamin B6 intake was negatively associated with the risk of all-cause and CVD mortality [31]. This is consistent with our current results and supports the finding that a range of high dietary vitamin B6 intake levels may have a beneficial effect on the risk of CVD death and all-cause mortality.

Pyridoxal 5′-phosphate (PLP) is the active coenzyme form of vitamin B6 [30]. PLP is involved in all metabolic processes relevant to CVD prevention (antioxidant activity, inflammatory processes, homocysteine catabolism, and phosphorylation-related reactions) [32]. The association between low vitamin B6 status and inflammation has been demonstrated in previous studies [33]. Vitamin B6 acts as an antioxidant by scavenging free radicals and reducing lipid peroxidation [34]. Excessive accumulation of oxidants or a lack of antioxidants in the body may result in oxidative stress, which in turn may cause vascular inflammation and endothelial disorders [30]. In a large prospective randomized trial, supplementation with vitamin B6, folic acid, and vitamin B12 was found to reduce the risk of stroke in patients with congenital heart disease by lowering homocysteine [35]. A meta-analysis also showed that a 3 mol/L reduction in homocysteine levels was associated with a 19% reduction in the risk of stroke and an 11% reduction in the risk of coronary heart disease [28].

Current recommendations only consider zinc intake but do not take into account the many dietary factors that influence zinc utilization. Previous studies have found that dietary intake of vitamin B6 affects the absorption of dietary zinc [14]. Therefore, we hypothesized that there is an interaction between dietary vitamin B6 and dietary zinc. In terms of CVD mortality, as expected, we observed a statistically significant interaction between dietary zinc and vitamin B6. In further analyses, higher dietary vitamin B6 and higher dietary zinc were associated with a lower risk of CVD mortality. In terms of all-cause mortality, we found that higher dietary vitamin B6 levels and higher dietary zinc levels were associated with a lower risk of all-cause mortality, but there was no interaction between dietary zinc and vitamin B6. In previous studies, it has been suggested that lower zinc bioavailability may be associated with a higher risk of atherosclerosis [36]. It has been hypothesized that pyridoxic acid, a metabolite of tryptophan, could facilitate the intestinal absorption of zinc. Pyridoxic acid derived from tryptophan has the action of the enzyme kynureninase, which is dependent on pyridoxine phosphate; therefore, adequate zinc absorption is indirectly dependent on an adequate supply of vitamin B6 [37]. It has also been shown that the increased fecal excretion of zinc and copper in animal populations deficient in vitamin B6 may due to a reduction in the absorption of these elements from the diet rather than excretion from tissue stores [38]. The positive combined effect found between dietary zinc and vitamin B6 could therefore be explained by the fact that higher dietary vitamin B6 facilitated the absorption of dietary zinc, thereby reducing the risk of mortality.

The range for intake of essential elements should be discussed within the framework of a model of health effects, as low intake levels of elements can lead to their deficiency, whereas high intake levels may cause toxicity [39]. If zinc intake is very high, signs of toxicity (vomiting, nausea, drowsiness and fatigue, epigastric pain) are evident [40]. The balance of essential elements is therefore crucial to the individual’s system of homeostasis, but the question of zinc utilization has not been considered in current recommendations [39]. The dietary zinc–vitamin B6 ratio may therefore be an important influential factor in cardiac mortality. By integrating and analyzing data from the large NHANES survey, we propose for the first time that the dietary zinc–vitamin B6 ratio is an influential factor in CVD mortality in US adults. To exclude the effects of dietary zinc and vitamin B6 intake, the model further adjusted for dietary zinc and vitamin B6 intake to show a J-shaped association between dietary zinc–vitamin B6 ratio and mortality risk, with a high dietary zinc–vitamin B6 ratio (high dietary zinc, low vitamin B6) significantly associated with an increased risk of CVD mortality. In our three study groups, a moderate dietary zinc–vitamin B6 ratio (range: 4.48–7.54) had the lowest risk of mortality.

In summary, dietary zinc and vitamin B6 intake levels are of concern. In practical terms, this study suggests that dietary supplementation with both zinc and vitamin B6 may be more beneficial in reducing the risk of cardiovascular disease, and that attention should be paid to the dietary zinc–vitamin B6 ratio when supplementing, as an excessive dietary zinc–vitamin B6 ratio may elevate the risk of cardiovascular disease, providing a basis for further research into the relationship between dietary zinc and vitamin B6 and cardiovascular disease.

The best sources of zinc include lamb, beef, fish, cereals, root vegetables, legumes, and nuts [25]. Foods rich in vitamin B6 include white meat, red meat, spinach, eggs, potatoes, bananas, nuts, and legumes [31]. However, the association of red and processed meat consumption with increased risk of all-cause mortality and CVD incidence has been consistently demonstrated in previous studies, with white meat showing a neutral association with CVD [41]. Increased intake of plant-based foods (whole grains, fruits, vegetables, legumes, and nuts) is associated with a reduced risk of atherosclerosis [42]. This may be related to the richness of red and unprocessed meat in saturated fatty acids and the high use of preservatives, such as sodium salts and nitrates, in the preparation of processed meat products [41]. However, some bioactive compounds (phytates, saponins, tannins) have been found to interfere with nutritional accessibility and intestinal uptake in legumes [43]. Vegetarians have lower zinc bioavailability compared to non-vegetarians due to the lack of meat intake and increased intake of phytate-containing legumes [44]. Therefore, supplementation with foods with high vitamin B6 utilization and low phytates (e.g., bananas) may improve zinc absorption and thus further reduce the risk of CVD. This finding needs to be explored in further studies [45].

Our prospective study has a number of strengths. To our knowledge, this study is one of the largest investigations into the association of dietary zinc and vitamin B6 levels with CVD death and all-cause mortality, adjusted for many potentially influential factors. In addition, the study has a nationally representative population, which facilitates the generalization of the findings. This study also has some limitations. First, although we adjusted for potential influencing factors, we could not completely exclude residual confounding effects caused by other variables, random errors, or the effect of uncontrolled factors, which may have biased the results to some extent. Secondly, dietary zinc intake levels and vitamin B6 intake levels were based on averages derived from two follow-up visits and may not accurately reflect long-term status. Finally, higher dietary zinc–vitamin B6 ratios or non-dual high combinations (high dietary zinc and high vitamin B6 combined) are associated with an increased risk of CVD mortality, which may be partly attributable to the inflammatory and oxidative stress caused by zinc deficiency and the role of dietary vitamin B6 in promoting dietary zinc absorption, as well as toxicity symptoms resulting from very high zinc intakes. The exact mechanisms need to be further analyzed in other studies.

## 5. Conclusions

Our current study shows that low dietary zinc levels and low dietary vitamin B6 levels are associated with an increased risk of CVD mortality. There was an interaction between dietary zinc and dietary vitamin B6, with the risk of CVD mortality being minimized in the group with high dietary zinc intake and high vitamin B6 intake. There was also a trend towards an increased risk of CVD mortality in people with high dietary zinc–vitamin B6 ratios after adjusting for dietary zinc intake and vitamin B6 intake. Nevertheless, more evidence is needed to determine the optimal range of dietary zinc levels, dietary vitamin B6 levels, and dietary zinc–vitamin B6 ratios to reduce the risk of CVD mortality.

## Figures and Tables

**Figure 1 nutrients-15-00420-f001:**
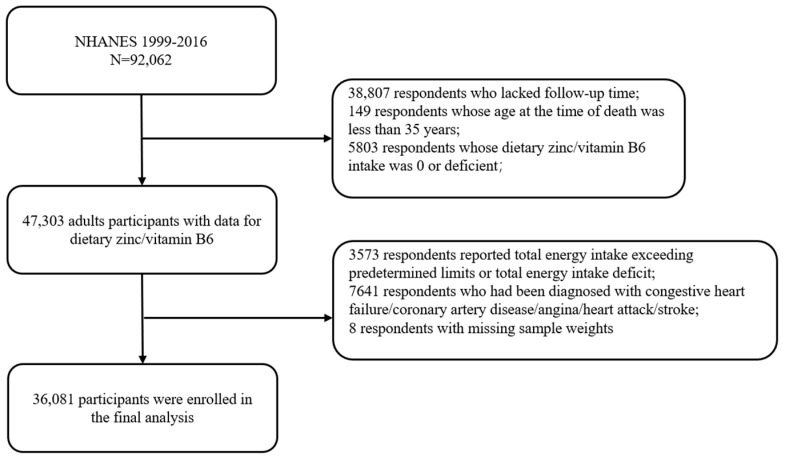
Complete exclusion flow chart for inclusion.

**Figure 2 nutrients-15-00420-f002:**
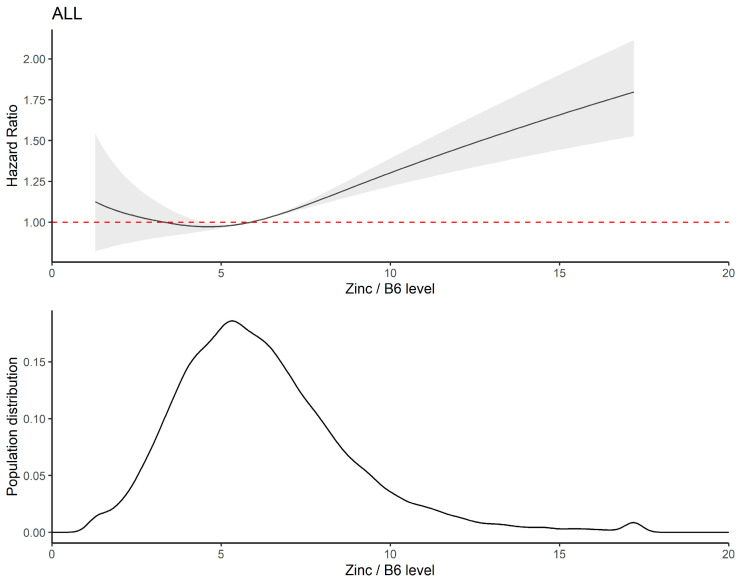
Association of dietary zinc–vitamin B6 ratio with CVD mortality in adults from NHANES, 1999–2016. In Figure 2, the red dotted line is Hazard Ratio Reference Line.

**Table 1 nutrients-15-00420-t001:** Characteristics of study subjects screened by median dietary zinc intake from the National Health and Nutrition Examination Survey (NHANES, 1999–2016) ^a^.

	Participants, *n* (%)					
	Zinc Level, mg/day			Vitamin B6 Level, mg/day		
Characteristics	<9.87	≥9.87	*p*	<1.73	≥1.73	*p*
Age, mean (SD), years	46.72 (17.07)	44.68 (15.66)	<0.001	45.98 (16.70)	45.32 (16.04)	0.005
Sex, %						
Male	5196 (31.1)	11,295 (58.3)	<0.001	5573 (32.6)	10,916 (57.5)	<0.001
Female	11,511 (68.9)	8079 (41.7)		11,523 (67.4)	8069 (42.5)	
BMI, %						
<30 kg/m^2^	11,010 (65.9)	12,923 (66.7)	0.323	11,027 (64.5)	12,891 (67.9)	<0.001
≥30 kg/m^2^	5697 (34.1)	6451 (33.3)		6069 (35.5)	6094 (32.1)	
Education level, %						
<High school	3169 (19.0)	2727 (14.1)	<0.001	3313 (19.4)	2579 (13.6)	<0.001
High school	4037 (24.2)	4256 (22)		4235 (24.8)	4077 (21.5)	
College or above	9458 (56.7)	12,361 (63.9)		9511 (55.7)	12,306 (64.9)	
Race/Ethnicity, %						
Mexican American	1320 (7.9)	1666 (8.6)	<0.001	1368 (8.0)	1614 (8.5)	<0.001
Non-Hispanic white	10,926 (65.4)	13,930 (71.9)		11,437 (66.9)	13,422 (70.7)	
Non-Hispanic black	2239 (13.4)	1686 (8.7)		2205 (12.9)	1728 (9.1)	
Other	2222 (13.3)	2092 (10.8)		2086 (12.2)	2221 (11.7)	
Family income–poverty ratio, %						
<1.0	2508 (16.3)	2105 (11.8)	<0.001	2583 (16.4)	2029 (11.6)	<0.001
1.0–3.0	5800 (37.7)	5780 (32.4)		5875 (37.3)	5685 (32.5)	
>3.0	7077 (46.0)	9973 (55.9)		7293 (46.3)	9760 (55.8)	
Smoking, %						
Never	8496 (50.9)	9504 (49.1)	<0.001	8352 (48.9)	9655 (50.9)	<0.001
Ever	3205 (19.2)	4413 (22.8)		3211 (18.8)	4419 (23.3)	
Currently	4991 (29.9)	5439 (28.1)		5534 (32.4)	4894 (25.8)	
Alcohol, %						
Yes	11,133 (70.8)	14,442 (79.2)	<0.001	11,392 (70.8)	14,170 (79.3)	<0.001
No	4592 (29.2)	3793 (20.8)		4699 (29.2)	3699 (20.7)	
HbA1c, %						
<7%	13,451 (95.4)	15,680 (95.9)	0.06	13,750 (95.3)	15,397 (96.1)	0.006
≥7%	649 (4.6)	670 (4.1)		678 (4.7)	625 (3.9)	
High blood pressure, %						
Yes	4808 (28.9)	4939 (25.6)	<0.001	4903 (28.8)	4859 (25.7)	<0.001
No	11,830 (71.1)	14,355 (74.4)		12,122 (71.2)	14,048 (74.3)	
High cholesterol, %						
Yes	5148 (34.9)	6089 (35.6)	0.409	5268 (34.9)	5984 (35.7)	0.246
No	9602 (65.1)	11,015 (64.4)		9826 (65.1)	10,777 (64.3)	

^a^ All data analyses in this study were corrected based on weighted estimates of the sample weights provided by NHANES. Continuous variables are expressed as means (standard errors). Categorical variables are expressed as *n* (%).*p*-values were calculated using *t*-tests for continuous variables and chi-square tests for categorical variables. Abbreviations: BMI, body mass index (calculated as weight in kilograms divided by squared height in m^2^); HbA1c, glycated hemoglobin.

**Table 2 nutrients-15-00420-t002:** Hazard ratios for CVD and all-cause mortality by dietary zinc and vitamin B6 intake levels in NHANES, 1999–2016.

	Hazard Ratio (95% CI)			
	Zinc Level, mg		B6 Level, mg	
	Low (<9.89 mg/day)	High (≥9.89 mg/day)	Low (<1.73 mg/day)	High (≥1.73 mg/day)
Diseases of heart mortality				
Deaths, No./Total No.	620/16,208	443/16,179	643/16,194	420/16,193
Model 1 ^a^	1.00 [Reference]	0.76 (0.73,0.79)	1.00 [Reference]	0.71 (0.63,0.80)
Model 2 ^b^	1.00 [Reference]	0.85 (0.83,0.87)	1.00 [Reference]	0.91 (0.86,0.96)
All-cause mortality				
Deaths, No./Total No.	2680/18,043	2077/18,038	2705/18,042	2052/18,039
Model 1 ^a^	1.00 [Reference]	0.91 (0.90,0.91)	1.00 [Reference]	0.84 (0.83,0.84)
Model 2 ^b^	1.00 [Reference]	0.93 (0.87,1.00)	1.00 [Reference]	0.91 (0.90,0.93)

^a^ Model 1 adjusted for sex (male or female), age (years). ^b^ Model 2 adjusted for sex (male or female), age (years), BMI (<30 or ≥30 kg/m^2^), education level (<high school, high school or equivalent, college and above), race, income–poverty ratio (<1.0, 1.0–3.0, >3.0), smoking status (yes or no), alcohol consumption status (yes or no), HbA1c (<7% or 7%), history of hypertension (yes or no), history of high cholesterol (yes or no).

**Table 3 nutrients-15-00420-t003:** Hazard ratios for CVD and all-cause mortality by different dose groups and levels of dietary zinc and vitamin B6 intake in NHANES, 1999–2016.

Dietary Zinc	Vitamin B6	HR (95% CI)	HR (95% CI)
		Diseases of Heart	All Cause
Model 1 ^a^			
Low	Low	1.56 (1.40,1.72)	1.21 (1.20,1.23)
Low	High	1.41 (1.32,1.50)	1.03 (0.95,1.11)
High	Low	1.59 (1.42,1.78)	1.19 (1.10,1.28)
High	High	1.00 [Reference]	1.00 [Reference]
*p* _Interaction_		<0.001	0.966
Model 2 ^b^			
Low	Low	1.21 (1.12,1.29)	1.12 (1.07,1.17)
Low	High	1.42 (1.34,1.50)	1.05 (1.00,1.10)
High	Low	1.28 (1.14,1.45)	1.09(0.98,1.23)
High	High	1.00 [Reference]	1.00 [Reference]
*p* _Interaction_		<0.001	0.559

^a^ Model 1 adjusted for sex (male or female), age (years). ^b^ Model 2 adjusted for sex (male or female), age (years), BMI (<30 or ≥30 kg/m^2^), education level (<high school, high school or equivalent, college and above), race, income–poverty ratio (<1.0, 1.0–3.0, >3.0), smoking status (yes or no), alcohol consumption status (yes or no), HbA1c (<7% or 7%), history of hypertension (yes or no), history of high cholesterol (yes or no). LZLV group (dietary zinc level < 9.89 mg/day, vitamin B6 level < 1.73 mg/day); LZHV group (dietary zinc level < 9.89 mg/day, vitamin B6 level ≥ 1.73 mg/day); HZLV group (dietary zinc level ≥ 9.89 mg/day, vitamin B6 level < 1.73 mg/day); HZHV group (dietary zinc level ≥ 9.89 mg/day, vitamin B6 level ≥ 1.73 mg/day).

**Table 4 nutrients-15-00420-t004:** Association of dietary zinc–vitamin B6 ratio with CVD and all-cause mortality in NHANES, 1999–2016.

	Hazard Ratio (95% CI)		
	Zinc/B6 Level		
	Low (<4.48)	Medium (4.48–7.54)	High (>7.54)
Diseases of heart mortality			
Deaths, No./Total No.	262/8074	511/16,140	289/8173
Model 1 ^a^	1.00 (0.80,1.25)	1.00 [Reference]	1.26 (1.22,1.30)
Model 2 ^b^	1.00 (0.68,1.46)	1.00 [Reference]	1.27 (1.19,1.35)

^a^ Model 1 adjusted for sex (male or female), age (years). ^b^ Model 2 adjusted for zinc intake (mg/day), vitamin B6 intake (mg/day), sex (male or female), age (years), BMI (<30 or ≥30 kg/m^2^), education level (<high school, high school or equivalent, college and above), race, income–poverty ratio (<1.0, 1.0–3.0, >3.0), smoking status (yes or no), alcohol consumption status (yes or no), HbA1c (<7% or 7%), history of hypertension (yes or no), history of high cholesterol (yes or no).

**Table 5 nutrients-15-00420-t005:** Sensitivity analysis: multivariable-adjusted hazard ratios for dietary zinc–vitamin B6 ratio and CVD in NHANES, 1999–2016 ^a^.

	Hazard Ratio (95% CI) ^b^		
	Zinc/B6 Level, mg		
	Low (<4.48)	Medium (4.48–7.54)	High (>7.54)
Main analysis	1.00 (0.68,1.46)	1.00 [Reference]	1.27 (1.19,1.35)
Died within 2 years	0.96 (0.62,1.49)	1.00 [Reference]	1.36 (1.25,1.49)
Changing allowable energy limits (percentiles 1–99)	1.01 (0.70,1.45)	1.00 [Reference]	1.33 (1.26,1.40)
Adjusted for history of diabetes mellitus	0.98 (0.69,1.39)	1.00 [Reference]	1.31 (1.24,1.39)
Adjusted for dietary fiber intake	0.96 (0.65,1.42)	1.00 [Reference]	1.24 (1.18,1.31)

^a^ Values are HR estimated by Cox regression and 95% CI. Results are not significant if CI includes 1.00; *p* > 0.05 (two-tailed). ^b^ Model adjusted for zinc intake (mg/day), vitamin B6 intake (mg/day), sex (male or female), age (years), BMI (<30 or ≥30 kg/m^2^), education level (<high school, high school or equivalent, college and above), race, income–poverty ratio (<1.0, 1.0–3.0, >3.0), smoking status (yes or no), alcohol consumption status (yes or no), HbA1c (<7% or 7%), history of hypertension (yes or no), history of high cholesterol (yes or no).

## Data Availability

All NHANES data for this study are publicly available and can be found here: https://wwwn.cdc.gov/nchs/nhanes (accessed on 11 December 2022).

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
