# Peer review of "Associations of Dietary Zinc–Vitamin B6 Ratio with All-Cause Mortality and Cardiovascular Disease Mortality Based on National Health and Nutrition Examination Survey 1999–2016"

_nutrients, 2023, doi:10.3390/nu15020420_

Round 1

Reviewer 1 Report

This study is highly interesting to readers as it provides a novel finding on the interaction between dietary zinc and vitamin B6 intake in the association with CVD and all-cause mortality.

A few minor things are needed for the authors' attention for the paper to be published.

1. Line 18-- Vitamin B6 "intake" level.

2. Line 45-- if Zinc is capitalized throughout the content, it needs to be done so.  

3. Line 82-- why this citation needs to be from the reference #17?  There has been available reference prior to this one (to be published in 2023).

4. Line 191-- the BMI and HbA1c need to be specified. (BMI > &=30), HbA1c > &=7% ??

5. Line 230--- HZLV group--  is this typo?  I believe it should be HZHV..

6. Line 238---  education, race (<high school, high school..)....  is this correct?  Similarly other footnotes need correction.

7. Line 285--- this large "perspective" cohort study--- should it be "prospective"?

8. Line 343--  to "prevent" the development---  This statement is overstated as this citation is from a case-control study which cannot provide a causal cause-effect relationship. 

Reviewer 2 Report

A highly interesting paper dealing with Zn and other dietary factors proposing synergistic effects of Zn  with Vit B6 on CVD.  This study might  stimulate further prospective intervention studies on combined treatments with both factors - but who will pay for it (no apparent commercial interests !)? Hence, authors should address dieticians to provide longterm diets rich in both factors . What´s e.g.  about vegetarians with lower bioavailability of Zn?

Reviewer 3 Report

The ms considers an important issue of associations of dietary zinc-vitamin B6 ratio with Mortality. Except for modifying and sending the ms to an expert in English. The ms can be published. 
